# Traumatic Pseudoaneurysms of the Internal Mammary Artery: Two Cases and Percutaneous Intervention

**DOI:** 10.3390/diagnostics14010063

**Published:** 2023-12-27

**Authors:** Kayla A. Aikins, Zoé N. Anderson, Timothy M. Koci

**Affiliations:** Reno School of Medicine, University of Nevada, Reno, NV 89557, USA; kaikins@med.unr.edu (K.A.A.); zoeanderson@med.unr.edu (Z.N.A.)

**Keywords:** internal mammary artery, internal thoracic artery, interventional radiology, pseudoaneurysm, CT, percutaneous intervention

## Abstract

Pseudoaneurysms involving the internal mammary artery/internal thoracic artery (IMA/ITA) are rare occurrences, and the presentation and treatment approaches for such cases can be variable. Due to the potentially life-threatening risk of rupture, leading to conditions like hemothorax, it is important to have a comprehensive understanding of safe and effective diagnostic and therapeutic techniques. We present two cases of IMA/ITA artery pseudoaneurysms. A 91-year-old male presented to the emergency department following a motor vehicle accident. A CT scan of the chest revealed an anterior mediastinal hemorrhage with active extravasation. Percutaneous intervention revealed a pseudoaneurysm arising from a left IMA/ITA side branch. Coil embolization effectively treated the pseudoaneurysm. In the second case, a 79-year-old male presented with a sternal fracture after a ground-level fall, with parasternal hematoma and active bleeding (pseudoaneurysm) on Trauma Computerized Tomography of the chest with contrast. He underwent coil embolization, and subsequent post-procedure angiograms confirmed the effective occlusion of the left IMA/ITA, with no further visualization of the pseudoaneurysm. These two cases underscore the importance of tailored approaches in treating internal mammary artery pseudoaneurysms.

## 1. Introduction

Blunt chest trauma is common, and thoracic fractures, hemothorax, pneumothorax, aortic intimal injury or even aortic transection, and pulmonary contusions are well known. In the context of pseudoaneurysms, particularly those related to the IMA/ITA, are a less common consequence of blunt chest trauma. Pseudoaneurysms are vascular abnormalities that may occur at the site of arterial injury. Pseudoaneurysms do not contain any layer of the vessel wall, unlike a true aneurysm. In the occurrence of a pseudoaneurysm, blood is contained by a wall that is formed by clotting cascade products [1]. Thus, a perfused sac that communicates with the arterial lumen is formed [2]. Femoral pseudoaneurysms following access for endovascular procedures represent the most frequent clinical presentation of a pseudoaneurysm [1].

Pseudoaneurysms of the IMA/ITA are exceptionally rare but potentially lethal. The etiologies comprise 28.8% attributed to sternotomies, 13.6% arising from central venous catheter placements, and 27.7% linked to blunt chest trauma [3]. They may also occur due to conditions like Ehlers-Danlos syndrome, Kawasaki disease, fibromuscular dysplasia, type 1 neurofibromatosis, and systemic lupus erythematosus [1].

Prompt diagnosis and intervention are crucial because these pseudoaneurysms can lead to severe complications, including significant bleeding, hemomediastinum, hemothorax, hemopneumothorax, pneumothorax, diaphragmatic palsy, and hemorrhagic shock.

## 2. Case Presentation

A 91-year-old-male with no significant past medical history presented following a motor vehicle accident. The patient, a rear seat restrained passenger, was involved in a front-end collision at 55 miles per hour. On admission, the patient complained of chest and abdominal pain. A seat belt shoulder strap injury was noted on the physical exam. In addition to a mediastinal hematoma, pulmonary contusion, sternal fracture, and multiple rib fractures, a CT scan of the chest showed active extravasation (Figure 1). Percutaneous intervention of the left internal mammary artery demonstrated a bilobed pseudoaneurysm arising from a LIMA side branch. Coil embolization was performed with the placement of detachable coils above, below, and across the origin of the side branch, the source of bleeding. Control percutaneous intervention after coil placement showed the cessation of extravasation and no further visualization of the pseudoaneurysm (Figure 2).

A 79-year-old male was admitted to the hospital with chest pain following a GLF. In the emergency department (ED), clinical evaluation and CT imaging revealed a sternal fracture, retrosternal hemorrhage, right pleural effusion, and left parasternal contrast extravasation. Interventional radiology was consulted, and percutaneous intervention demonstrated that a pseudoaneurysm, measuring approximately 16 mm in size, was located caudad to the clavicular head arising from a left internal mammary artery side branch (Figure 3). The patient underwent successful coil embolization, and post-procedure angiograms confirmed the effective occlusion of the left internal mammary artery, with no further visualization of the pseudoaneurysm (Figure 4). A final left subclavian control angiogram demonstrated the absence of any collateral filling of the pseudoaneurysm. Despite the successful treatment of the aneurysm, severe hepatic encephalopathy due to chronic alcoholic liver disease severely complicated recovery. Following extensive medical care, a decision was made to transition the patient to hospice care.

91-year-old male:

**Figure 1 diagnostics-14-00063-f001:**
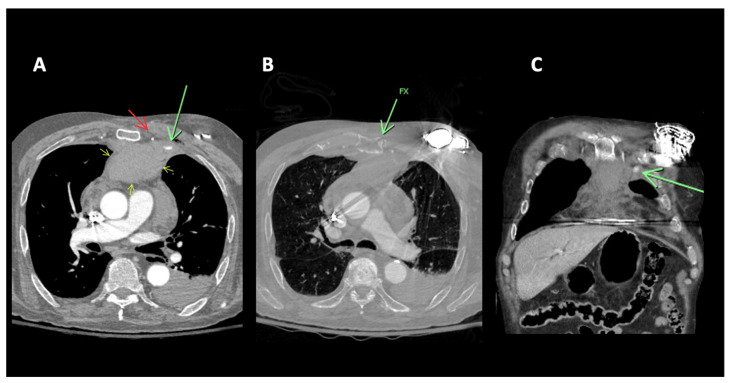
91M MVA restrained rear seat passenger. (**A**) CT of the chest with contrast. Focal contrast extravasation consistent with active bleeding site left parasternal anterior mediastinum (green arrow). Left internal mammary artery (red arrow). Anterior mediastinal hematoma (yellow arrows). (**B**) CT of the chest, bone windows. Sternal fracture (green arrow). (**C**) CT of the chest with contrast coronal reconstruction. Focal contrast extravasation consistent with active bleeding site left parasternal anterior mediastinum (green arrow).

**Figure 2 diagnostics-14-00063-f002:**
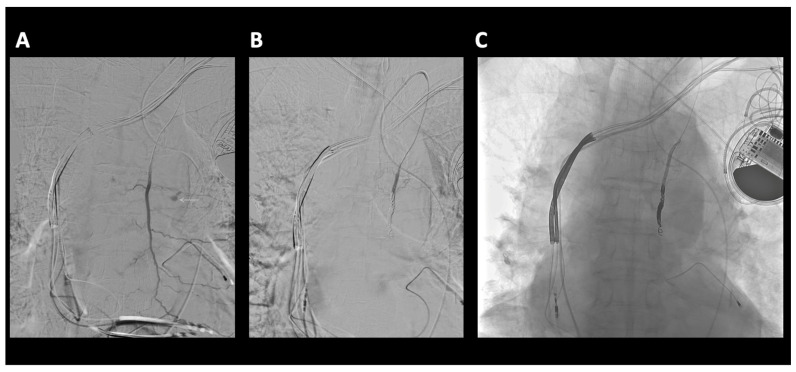
(**A**) Selective microcatheter digital subtraction angiogram (DSA) left internal mammary artery (LIMA) extravasation consistent with active bleeding/pseudoaneurysm from side branch of LIMA (white arrow). (**B**) Microcatheter DSA LIMA post coil embolization. Coils are deposited as a bridge covering the side branch bleeding source, sometimes referred to as sandwich technique or “jailing technique”, to eliminate antegrade as well as any retrograde or collateral flow to the bleeding side branch. No further extravasation. (**C**) Microcatheter “native” non-subtracted digital angiogram better demonstrates the occlusive coil pack.

79-year-old male:

**Figure 3 diagnostics-14-00063-f003:**
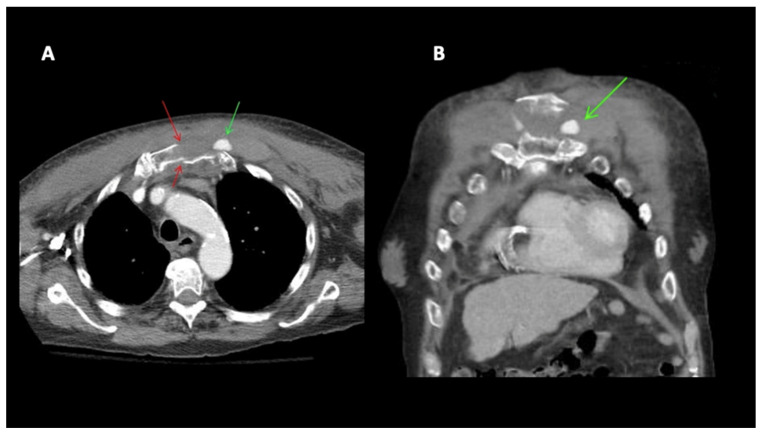
(**A**) 79M. Fall. CT chest with contrast. Sternal fracture (red arrows). Left parasternal active extravasation of contrast/pseudoaneurysm (green arrow). (**B**) Chest CT. Coronal reconstruction. Left parasternal active extravasation of contrast (green arrow).

**Figure 4 diagnostics-14-00063-f004:**
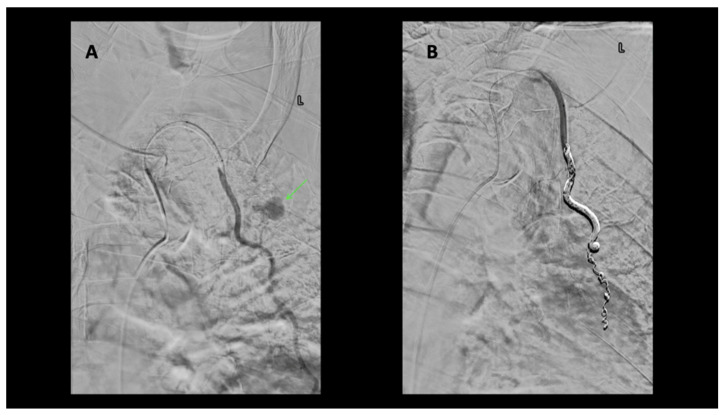
(**A**) Left internal mammary artery (LIMA) microcatheter DSA angiogram. Left parasternal active extravasation/pseudoaneurysm from side branch of LIMA (green arrow). (**B**) Microcatheter DSA LIMA post coil embolization. Coils are deposited as a bridge covering the side branch bleeding source, sometimes referred to as sandwich technique or “jailing technique”, to eliminate antegrade as well as any retrograde or collateral flow to the bleeding side branch. No further extravasation.

## 3. Results & Discussion

The internal mammary artery, also known as the internal thoracic artery, arises from the subclavian artery and travels along the inner surface of the chest wall to supply the anterior chest wall and breast. It is located roughly 1 to 2 cm lateral to the sternal margin on either side, between the transversus thoracis muscle posteriorly and the internal intercostal muscles and costal cartilages anteriorly [4]. As it descends, it gives rise to anterior intercostal arteries at each intercostal space and perforating cutaneous branches. At the sixth to seventh costal cartilages, it bifurcates into the musculophrenic artery and superior epigastric artery. This anatomical course makes the IMA vulnerable to injury [5].

Pseudoaneurysms of the IMA/ITA result from thoracic trauma or medical procedures. For example, numerous cases of pseudoaneurysm development have been reported following blunt chest trauma [6,7,8]. Less common causes include fibromuscular dysplasia, polyarteritis nodosa, systemic lupus erythematosus, and connective tissue diseases [2]. Signs and symptoms may include chest pain, cough, dyspnea, hemoptysis, hematoma, mass, murmur, vibration, and painful swelling. Complications of bleeding can lead to hemomediastinum, with or without extrinsic cardiac tamponade, hemothorax, hemopneumothorax, pneumothorax, diaphragmatic dysfunction, and hemorrhagic shock.

The diagnosis of an IMA/ITA pseudoaneurysm may necessiate chest CT, CTA of the chest, percutaneous intervention, or a combination of these imaging studies. Associated findings may include retrosternal/anterior mediastinal hematoma, a widened mediastinum, and an elevated diaphragm. In a report detailing an unusual cause of internal mammary pseudoaneurysm due to fibromuscular dysplasia, percutaneous intervention revealed a fusiform “string of beads” appearance [2]. Understanding the anatomy of the IMA/ITA and potential pathologies is crucial to ensure early detection and appropriate management.

The treatment of pseudoaneurysms primarily focuses on tailored interventions. Embolization is a minimally invasive procedure, involving the insertion of a catheter through a small incision, resulting in short recovery times and diminished postoperative complications. These interventions can be technically challenging but have a decreased risk of hemorrhage. The targeted nature of the procedure allows for the precise occlusion of the pseudoaneurysm, minimizing tissue damage and arterial compromise. Percutaneous intervention with embolization or stent-graft placement has led to a marked decrease in morbidity and mortality rates for pseudoaneurysms [9].

While percutaneous intervention with coil embolization is a primary consideration, sternotomy emerges as a valuable alternative, especially in cases where embolization is contraindicated. Contributing to this discourse, Noh et al. highlight sternotomy’s use in a critical scenario involving a 45-year-old male with active bleeding and extra-pericardial cardiac tamponade from a ruptured left IMA/ITA [10]. Regardless, sternotomy involves a higher rate of tissue disruption, cardiopulmonary complications, infection risk, blood loss, and an extended hospital stay.

In summary, this case emphasizes the need for additional research and collaboration to develop guidelines for effective intervention in these uncommon but potentially serious situations. The treatment of IMA/ITA pseudoaneurysms is essential to refine therapeutic approaches and address existing uncertainties. Current interventions, such as percutaneous techniques and sternotomy, lack comprehensive evidence on long-term outcomes and patient-specific factors influencing treatment decisions. Research efforts should focus on comparative effectiveness, novel technologies, and predictive factors for pseudoaneurysm formation in the IMA/ITA. Collaborative studies and systematic reviews can contribute to evidence-based practices, optimizing patient care and treatment algorithms for this vascular pathology.

## Data Availability

Data are contained within this article.

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
