# Peer review of "Traumatic Pseudoaneurysms of the Internal Mammary Artery: Two Cases and Percutaneous Intervention"

_diagnostics, 2023, doi:10.3390/diagnostics14010063_

Round 1

Reviewer 1 Report

Comments and Suggestions for Authors

General comments

In their paper entitled " Traumatic Pseudoaneurysms of the Internal Mammary Artery: Two Cases and Angiographic Intervention" authors describe their experience with 2 different cases of Internal Mammary artery embolization.

Specific comments

1) The cases are well written, with good imaging support.

2) The discussion is adequate.

3) Since the official name of the artery is nowadays internal thoracic artery, both names should be included in parentheses and/or initials (i.e. IMA/ITA).

4) It would be more appropriate to exchange the term "Angiographic" to the more precise term "Percutaneous Intervention".

5) ITA is an artery with tricky catheterization and reference to the catheter and microcatheter, as well as the brand names of the coils used might be difficult to our readers.

6) Overall, the cases are well presented, although they lack any novelty and would only be of interest to our less experienced readers.

Reviewer 2 Report

Comments and Suggestions for Authors

This manuscript is interesting case report regarding” Traumatic Pseudoaneurysms of the Internal Mammary Artery: 2 Two Cases and Angiographic Intervention”. I have some comments for the authors as follows:

Abstract section

-          In general, it is not necessary to include specific references in the abstract section of a manuscript. The purpose of the abstract is to provide a concise summary of this 2-case report.

-          It is advisable not to include the abbreviation in the abstract section if it is only mentioned once. However, it is recommended to use the full form of the abbreviation in the other parts of the manuscript. This ensures clarity and consistency throughout your work.

Introduction section

-          While the introduction briefly mentions that pseudoaneurysms are vascular abnormalities that occur at the site of arterial injury, it could benefit from providing more background information about the etiology, prevalence, and potential complications of pseudoaneurysms in general. This would help set the stage for understanding the specific focus on internal mammary artery pseudoaneurysms.

Discussion section

-          The discussion section primarily presents information about the internal mammary artery, its vulnerability to injury, and the causes and complications of internal mammary artery pseudoaneurysms. Please compare or analyze these findings in the context of existing previous literature or studies in order to strengthen the discussion.

-          Please provide more details regarding comprehensive analysis of the effectiveness, advantages, and potential complications associated with these interventions.

-          Please add the last paragraph for additional research and collaboration to develop guidelines for effective intervention in uncommon situations involving internal mammary artery pseudoaneurysms in order to provide any specific suggestions or directions for future research.
